# *ZeroFlow*: Scalable Scene Flow via Distillation

Kyle Vedder[1]*   Neehar Peri[2]   Nathaniel Chodosh[2]   Ishan Khatri[3]   Eric Eaton[1]
Dinesh Jayaraman[1]   Yang Liu[4]   Deva Ramanan[2]   James Hays[5]
[1]University of Pennsylvania   [2]Carnegie Mellon University   [3]Motional
[4]Lawrence Livermore National Laboratory   [5]Georgia Tech

## Abstract

Scene flow estimation is the task of describing the 3D motion field between temporally successive point clouds. State-of-the-art methods use strong priors and test-time optimization techniques, but require on the order of tens of seconds to process full-size point clouds, making them unusable as computer vision primitives for real-time applications such as open world object detection. Feedforward methods are considerably faster, running on the order of tens to hundreds of milliseconds for full-size point clouds, but require expensive human supervision. To address both limitations, we propose *Scene Flow via Distillation*, a simple, scalable distillation framework that uses a label-free optimization method to produce pseudo-labels to supervise a feedforward model. Our instantiation of this framework, *ZeroFlow*, achieves **state-of-the-art** performance on the *Argoverse 2 Self-Supervised Scene Flow Challenge* while using zero human labels by simply training on large-scale, diverse unlabeled data. At test-time, ZeroFlow is over $1000\times$ faster than label-free state-of-the-art optimization-based methods on full-size point clouds (34 FPS vs 0.028 FPS) and over $1000\times$ cheaper to train on unlabeled data compared to the cost of human annotation ($394 vs $\sim$750,000). To facilitate further research, we release our code, trained model weights, and high quality pseudo-labels for the Argoverse 2 and Waymo Open datasets at `https://vedder.io/zeroflow`.

## 1 Introduction

Scene flow estimation is an important primitive for open-world object detection and tracking (Najibi et al., 2022; Zhai et al., 2020; Baur et al., 2021; Huang et al., 2022; Erçelik et al., 2022). As an example, Najibi et al. (2022) generates supervisory boxes for an open-world LiDAR detector via offline object extraction using high quality scene flow estimates from Neural Scene Flow Prior (NSFP) (Li et al., 2021b). Although NSFP does not require human supervision, it takes tens of seconds to run on a single full-size point cloud pair. If NSFP were both high quality and real-time, its estimations could be directly used as a runtime primitive in the downstream detector instead of relegated to an offline pipeline. This runtime feature formulation is similar to Zhai et al. (2020)'s use of scene flow from FlowNet3D (Liu et al., 2019) as an input primitive for their multi-object tracking pipeline; although FlowNet3D is fast enough for online processing of subsampled point clouds, its supervised feedforward formulation requires significant in-domain human annotations.

Broadly, these exemplar methods are representative of the strengths and weakness of their class of approach. Supervised feedforward methods use human annotations which are expensive to annotate[1]. To amortize these costs, human annotations are typically done on consecutive observations, severely limiting the structural diversity of the annotated scenes (e.g. a 15 second sequence from an Autonomous Vehicle typically only covers a single city block); due to costs and labeling difficulty, large-scale labels are also rarely even available outside of Autonomous Vehicle domains. Test-time optimization techniques circumvent the need for human labels by relying on hand-built priors, but they are too slow for online scene flow estimation[2].

---

*Corresponding email: `kvedder@seas.upenn.edu`

[1]At $\sim$\$0.10 / cuboid / frame, the Argoverse 2 (Wilson et al., 2021) *train* split cost $\sim$\$750,000 to label; ZeroFlow's pseudo-labels cost \$394 at current cloud compute prices. See Supplemental E for details.

[2]NSFP (Li et al., 2021b) takes more than 26 seconds and Chodosh (Chodosh et al., 2023) takes more than 35 seconds per point cloud pair on the Argoverse 2 (Wilson et al., 2021) train split. See Supplemental E for details.

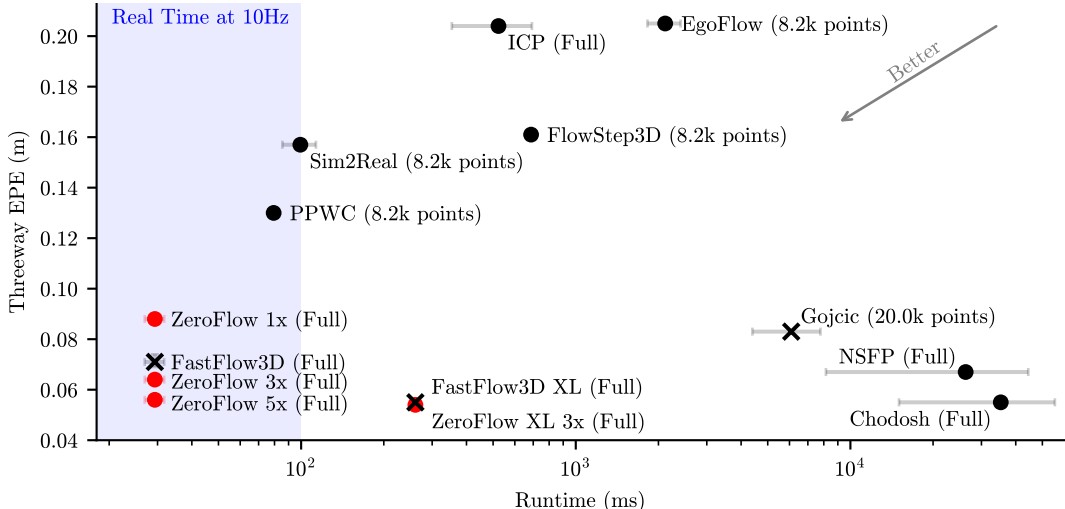

Figure 1: We plot the error and run-time of recent scene flow methods on the Argoverse 2 Sensor dataset (Wilson et al., 2021), along with the size of the point cloud prescribed in the method's evaluation protocol. Our method ZeroFlow 3X (ZeroFlow trained on $3\times$ pseudo-labeled data) outperforms its teacher (NSFP, Li et al. (2021b)) while running over $1000\times$ faster, and ZeroFlow XL 3X (ZeroFlow with a larger backbone trained on $3\times$ pseudo-labeled data) achieves **state-of-the-art**. Methods that use *any* human labels are plotted with ✗ , and zero-label methods are plotted with ●.

We propose *Scene Flow via Distillation* (SFvD), a simple, scalable distillation framework that uses a label-free optimization method to produce pseudo-labels to supervise a feedforward model. SFvD generates a new class of scene flow estimation methods that combine the strengths of optimization-based and feedforward methods with the power of data scale and diversity to achieve fast run-time and superior accuracy without human supervision. We instantiate this pipeline into *Zero-Label Scalable Scene Flow* (ZeroFlow), a family of methods that, motivated by real-world applications, can process full-size point clouds while providing high quality scene flow estimates. We demonstrate the strength of ZeroFlow on Argoverse 2 (Wilson et al., 2021) and Waymo Open (Sun et al., 2020), notably achieving **state-of-the-art** on the *Argoverse 2 Self-Supervised Scene Flow Challenge* (Figure 1).

Our primary contributions include:

- We introduce a simple yet effective distillation framework, *Scene Flow via Distillation* (SFvD), which uses a label-free optimization method to produce pseudo-labels to supervise a feedforward model, allowing us to surpass the performance of slow optimization-based approaches at the speed of feedforward methods.

- Using SFvD, we present *Zero-Label Scalable Scene Flow* (ZeroFlow), a family of methods that produce fast, **state-of-the-art** scene flow on full-size clouds, with methods running over $1000\times$ faster than state-of-the-art optimization methods (29.33 ms for ZeroFlow 1X vs 35,281.4 ms for Chodosh) on real point clouds, while being over $1000\times$ cheaper to train compared to the cost of human annotations ($394 vs ~$750,000).

- We release high quality flow pseudo-labels (representing 7.1 GPU months of compute) for the popular Argoverse 2 (Wilson et al., 2021) and Waymo Open (Sun et al., 2020) autonomous vehicle datasets, alongside our code and trained model weights, to facilitate further research.

## 2 BACKGROUND AND RELATED WORK

Given point clouds $P_t$ at time $t$ and $P_{t+1}$ at time $t + 1$, scene flow estimators predict $\hat{F}_{t,t+1}$, a 3D vector for each point in $P_t$ that describes how it moved from $t$ to $t + 1$ (Dewan et al., 2016). Performance is traditionally measured using the Endpoint Error (EPE) between the predicted flow

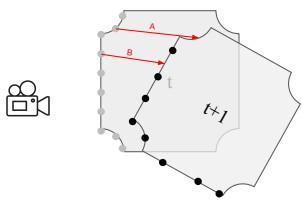

Figure 2: Scene Flow vectors describe where the point on an object at time $t$ will end up on the object at $t + 1$. In this example, ground truth flow vector *A*, associated with a point in the upper left concave corner of the object at $t$ has no nearby observations at $t + 1$ due to occlusion of the concave feature. The ground truth flow vector *B*, associated with a point on the face of the object at $t$, does not directly match with any observed point on the object at $t + 1$ due to observational sparsity. Thus, point matching between $t$ and $t + 1$ alone is insufficient to generate ground truth flow.

$\hat{F}_{t,t+1}$ and ground truth flow $F^*_{t,t+1}$ (Equation 1):

$$\text{EPE}\,(P_t) = \frac{1}{\|P_t\|} \sum_{p \in P_t} \left\| \hat{F}_{t,t+1}(p) - F^*_{t,t+1}(p) \right\|_2. \tag{1}$$

Unlike next token prediction in language (Radford et al., 2018) or next frame prediction in vision (Weng et al., 2021), future observations do not provide ground truth scene flow (Figure 2). To simply evaluate scene flow estimates, ground truth motion descriptions must be provided by an oracle, typically human annotation of real data (Sun et al., 2020; Wilson et al., 2021) or the generator of synthetic datasets (Mayer et al., 2016; Zheng et al., 2023).

Recent scene flow estimation methods either train feedforward methods via supervision from human annotations (Liu et al., 2019; Behl et al., 2019; Tishchenko et al., 2020; Kittenplon et al., 2021; Wu et al., 2020; Puy et al., 2020; Li et al., 2021a; Jund et al., 2021; Gu et al., 2019; Battrawy et al., 2022; Wang et al., 2022), perform human-designed test-time surrogate objective optimization over hand-designed representations (Pontes et al., 2020; Eisenberger et al., 2020; Li et al., 2021b; Chodosh et al., 2023), or learn from self-supervision from human-designed surrogate objectives (Mittal et al., 2020; Baur et al., 2021; Gojcic et al., 2021; Dong et al., 2022; Li et al., 2022).

Supervised feedforward methods are efficient at test-time; however, they require costly human annotations at train-time. Both test-time optimization and self-supervised feedforward methods seek to address this problem by optimizing or learning against label-free surrogate objectives, e.g. Chamfer distance (Pontes et al., 2020), cycle-consistency (Mittal et al., 2020), and various hand-designed rigidity priors (Dewan et al., 2016; Pontes et al., 2020; Li et al., 2022; Chodosh et al., 2023; Baur et al., 2021; Gojcic et al., 2021). Self-supervised methods achieve faster inference by forgoing expensive test-time optimization, but do not match the quality of optimization-based methods (Chodosh et al., 2023) and tend to require human-designed priors via more sophisticated network architectures compared to supervised methods (Baur et al., 2021; Gojcic et al., 2021; Kittenplon et al., 2021). In practice, this makes them slower and more difficult to train. In contrast to existing work, we take advantage of the quality of optimization-based methods as well as the efficiency and architectural simplicity of supervised networks. Our approach, ZeroFlow, uses label-free optimization methods (Li et al., 2021b) to produce pseudo-labels to supervise a feedforward model (Jund et al., 2021), similar to methods used for distillation in other domains Yang et al. (2023).

## 3 METHOD

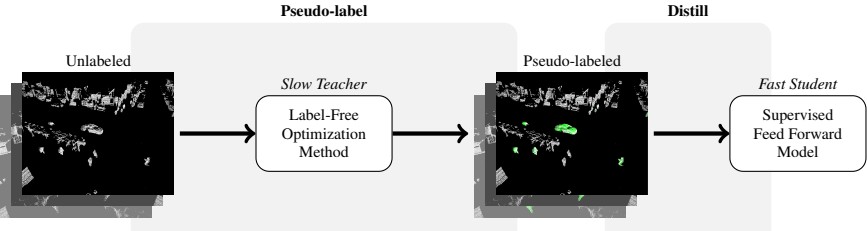

Figure 3: The *Scene Flow via Distillation* (SFvD) framework, which describes a new class of scene flow methods that produce high quality, human label-free flow at the speed of feedforward networks.

We propose *Scene Flow via Distillation* (SFvD), a simple, scalable distillation framework that creates a new class of scene flow estimators by using a label-free optimization method to produce pseudo-labels to supervise a feedforward model (Figure 3). While conceptually simple, efficiently instantiating SFvD requires careful construction; most online optimization methods and feedforward architectures are unable to efficiently scale to full-size point clouds (Section 3.1).

Based on our scalability analysis, we propose *Zero-Label Scalable Scene Flow* (ZeroFlow), a family of scene flow models based on SFvD that produces fast, **state-of-the-art** scene flow estimates for full-size point clouds without any human labels (Algorithm 1). ZeroFlow uses Neural Scene Flow prior (NSFP) (Li et al., 2021b) to generate high quality, label-free pseudo-labels on full-size point clouds (Section 3.2) and FastFlow3D (Jund et al., 2021) for efficient inference (Section 3.3).

## 3.1 SCALING SCENE FLOW VIA DISTILLATION TO LARGE POINT CLOUDS

Popular AV datasets including Argoverse 2 (Wilson et al. (2021), collected with dual Velodyne VLP-32 sensors) and Waymo Open (Sun et al. (2020), collected with a proprietary lidar sensor and subsampled) have full-size point clouds with an average of 52,000 and 79,000 points per frame, respectively, after ground plane removal (Supplemental A, Figure 6). For practical applications, sensors such as the Velodyne VLP-128 in dual return mode produce up to 480,000 points per sweep (Vel, 2019) and proprietary sensors at full resolution can produce well over 1 million points per sweep. Thus, scene flow methods must be able to process many points in real-world applications.

Unfortunately, most existing methods focus strictly on scene flow *quality* for toy-sized point clouds, constructed by randomly subsampling full point clouds down to 8,192 points (Jin et al., 2022; Tishchenko et al., 2020; Wu et al., 2020; Kittenplon et al., 2021; Liu et al., 2019; Li et al., 2021b). As we are motivated by real-world applications, we instead target scene flow estimation for the full-sized point cloud, making architectural efficiency of paramount importance. As an example of stark differences between feedforward architectures, FastFlow3D (Jund et al., 2021), which uses a PointPillar-style encoder (Lang et al., 2019), can process 1 million points in under 100 ms on an NVIDIA Tesla P1000 GPU (making it real-time for a 10Hz LiDAR), while methods like FlowNet3D (Liu et al., 2019) take almost 4 seconds to process the same point cloud.

We design our approach to efficiently process full-size point clouds. For SFvD's pseudo-labeling step, speed is less of a concern; pseudo-labeling each point cloud pair is offline and highly parallelizable. High-quality methods like Neural Scene Flow Prior (NSFP, Li et al. (2021b)) require only a modest amount of GPU memory (under 3GB) when estimating scene flow on point clouds with 70K points, enabling fast and low-cost pseudo-labeling using a cluster of commodity GPUs; as an example, pseudo-labeling the Argoverse 2 train split with NSFP is over $1000\times$ cheaper than human annotation (Supplemental E). The efficiency of SFvD's student feedforward model *is* critical, as it determines both the method's test-time speed and its training speed (faster training enables scaling to larger datasets), motivating models that can efficiently process full-size point clouds.

## 3.2 NEURAL SCENE FLOW PRIOR IS A SLOW TEACHER

Neural Scene Flow Prior (NSFP, Li et al. (2021b)) is an optimization-based approach to scene flow estimation. Notably, it does not use ground truth labels to generate high quality flows, instead relying upon strong priors in its learnable function class (determined by the coordinate network's architecture) and optimization objective (Equation 2). Point residuals are fit per point cloud pair $P_t$, $P_{t+1}$ at test-time by randomly initializing two MLPs; one to describe the forward flow $\hat{F}^+$ from $P_t$ to $P_{t+1}$, and one to describe the reverse flow $\hat{F}^-$ from $P_t + \hat{F}_{t,t+1}$ to $P_t$ in order to impose cycle consistency. The forward flow $\hat{F}^+$ and backward flow $\hat{F}^-$ are optimized jointly to minimize

$$\text{TruncatedChamfer}(P_t + \hat{F}^+, P_{t+1}) + \text{TruncatedChamfer}(P_t + \hat{F}^+ + \hat{F}^-, P_t) \ , \qquad (2)$$

where TruncatedChamfer is the standard Chamfer distance with per-point distances above 2 meters set to zero to reduce the influence of outliers.

NSFP is able to produce high-quality scene flow estimations due to its choice of coordinate network architecture and use of cycle consistency constraint. The coordinate network's learnable function class is expressive enough to fit the low frequency signal of residuals for moving objects while restrictive enough to avoid fitting the high frequency noise from TruncatedChamfer, and the cycle

consistency constraint acts as a local smoothness regularizer for the forward flow, as any shattering effects in the forward flow are penalized by the backwards flow. NSFP provides high quality estimates on full-size point clouds (Figure 1), so we select NSFP for ZeroFlow's pseudo-label step of SFvD.

### 3.3 FASTFLOW3D IS A FAST STUDENT

FastFlow3D (Jund et al., 2021) is an efficient feedforward method that learns using human supervisory labels $F_{t,t+1}^*$ and per-point foreground / background class labels. FastFlow3D's loss minimizes a variation of the End-Point Error (Equation 1) that reduces the importance of annotated background points, thus minimizing

$$\frac{1}{\|P_t\|} \sum_{p \in P_t} \sigma(p) \left\| \hat{F}_{t,t+1}(p) - F_{t,t+1}^*(p) \right\|_2 \quad (3) \quad \text{where} \quad \sigma(p) = \begin{cases} 1 & \text{if } p \in \text{Foreground} \\ 0.1 & \text{if } p \in \text{Background} \end{cases}. \quad (4)$$

FastFlow3D's architecture is a PointPillars-style encoder (Lang et al., 2019), traditionally used for efficient LiDAR object detection (Vedder & Eaton, 2022), that converts the point cloud into a birds-eye-view pseudoimage using infinitely tall voxels (pillars). This pseudoimage is then processed with a 4 layer U-Net style backbone. The encoder of the U-Net processes the $P_t$ and $P_{t+1}$ pseudoimage separately, and the decoder jointly processes both pseudoimages. A small MLP is used to decode flow for each point in $P_t$ using the point's coordinate and its associated pseudoimage feature.

As discussed in Section 3.1, FastFlow3D's architectural design choices make fast even on full-size point clouds. While most feedforward methods are evaluated using a standard toy evaluation protocol with subsampled point clouds, FastFlow3D is able to scale up to full resolution point clouds while maintaining real-time performance and emitting competitive quality scene flow estimates using human supervision, making it a good candidate for the distillation step of SFvD.

In order to train FastFlow3D using pseudo-labels, we replace the foreground / background scaling function (Equation 4) with a simple uniform weighting ($\sigma(\cdot) = 1$), which collapses to Average EPE; see Supplemental B for experiments with other weighting schemes. Additionally, we depart from FastFlow3D's problem setup in two minor ways: we delete ground points using dataset provided maps, a standard pre-processing step (Chodosh et al., 2023), and use the standard scene flow problem setup of predicting flow between two frames (Section 2) instead of predicting future flow vectors in meters per second. Algorithm 1 describes our approach, with details specified in Section 4.1.

In order to take advantage of the unlabeled data scaling of SFvD, we expand FastFlow3D to a family of models by designing a higher capacity backbone, producing *FastFlow3D XL*. This larger backbone halves the size of each pillar to quadruple the pseudoimage area, doubles the size of the pillar embedding, and adds an additional layer to maintain the network's receptive field in metric space; as a result, the total parameter count increases from 6.8 million to 110 million.

---

**Algorithm 1** ZeroFlow

---

1:   $D \leftarrow$ collection of unlabeled point cloud pairs                  ▷ Training Data
2: **for** $P_t, P_{t+1} \in D$ **do**                             ▷ Parallel For
3:      $F_{t,t+1}^* \leftarrow \text{TeacherNSFP}(P_t, P_{t+1})$           ▷ SFvD *Pseudo-label* Step

4: **for** epoch $\in$ epochs **do**
5:      **for** $P_t, P_{t+1}, F_{t,t+1}^* \in D$ **do**                 ▷ SFvD's *Distill* Step
6:          $l \leftarrow \text{Equation 3}(\text{StudentFastFlow3D}_\theta(P_t, P_{t+1}), F_{t,t+1}^*)$
7:          $\theta \leftarrow \theta$ updated w.r.t. $l$

---

## 4 EXPERIMENTS

ZeroFlow provides a family of fast, high quality scene flow estimators. In order to validate this family and understand the impact of components in the underlying Scene Flow via Distillation framework, we perform extensive experiments on the Argoverse 2 (Wilson et al., 2021) and Waymo Open (Sun et al., 2020) datasets. We compare to author implementations of NSFP (Li et al., 2021b) and Chodosh et al. (2023), implement FastFlow3D (Jund et al., 2021) ourselves (no author implementation is available), and use Chodosh et al. (2023)'s implementations for all other baselines.

As discussed in Chodosh et al. (2023), downstream applications typically rely on good quality scene flow estimates for foreground points. Most scene flow methods are evaluated using average Endpoint Error (EPE, Equation 1); however, roughly 80% of real-world point clouds are background, causing average EPE to be dominated by background point performance. To address this, we use the improved evaluation metric proposed by Chodosh et al. (2023), *Threeway EPE*:

$$\text{Threeway EPE}(P_t) = \text{Avg} \begin{cases} \text{EPE}(p \in P_t : p \in \text{Background}) & \text{(Static BG)} \\ \text{EPE}(p \in P_t : p \in \text{Foreground} \land F^*_{t,t+1}(p) \leq 0.5\text{m/s}) & \text{(Static FG)} \\ \text{EPE}(p \in P_t : p \in \text{Foreground} \land F^*_{t,t+1}(p) > 0.5\text{m/s}) & \text{(Dynamic FG)} \end{cases} \quad (5)$$

## 4.1 HOW DOES ZEROFLOW PERFORM COMPARED TO PRIOR ART ON REAL POINT CLOUDS?

The overarching promise of ZeroFlow is the ability to build fast, high quality scene flow estimators that improve with the the availability of large-scale *unlabeled* data. Does ZeroFlow deliver on this promise? How does it compare to state-of-the-art methods?

To characterize the ZeroFlow family's performance, we use Argoverse 2 to perform scaling experiments along two axes: dataset size and student size. For our standard size configuration, we use the Argoverse 2 Sensor *train* split and the standard FastFlow3D architecture, enabling head-to-head comparisons against the fully supervised FastFlow3D as well as other baseline methods. For our scaled up dataset (denoted *3X* in all experiments), we use the Argoverse 2 Sensor *train* split and concatenate a roughly twice as large set of unannotated frame pairs from the Argoverse 2 LiDAR dataset, uniformly sampled from its 20,000 sequences to maximize data diversity. For our scaled up student architecture (denoted *XL* in all experiments), we use the XL backbone described in Section 3.3. For details on the exact dataset construction and method hyperparameters, see Supplemental A

Table 1: Quantitative results on the Argoverse 2 Sensor validation split using the evaluation protocol from Chodosh et al. (2023). The methods used in this paper, shown in the first two blocks of the table, are trained and evaluated on point clouds within a 102.4m × 102.4m area centered around the ego vehicle (the settings for the *Argoverse 2 Self-Supervised Scene Flow Challenge*) . However, following the protocol of Chodosh et al. (2023), all methods report error on points in the 70m × 70m area centered around the ego vehicle. Runtimes are collected on an NVIDIA V100 with a batch size of 1 (Peri et al., 2023). FastFlow3D, ZeroFlow 1X, and ZeroFlow 3X have identical feedforward architectures and thus share the same real-time runtime; FastFlow3D XL, ZeroFlow XL 1X, and ZeroFlow XL 3X have identical feedforward architectures and thus share the same runtime. Methods with an * have performance averaged over 3 training runs (see Supplemental C for details). Underlined methods require human supervision.

| | Runtime (ms) | | Point Cloud Subsampled Size | Threeway EPE | Dynamic FG EPE | Static FG EPE | Static BG EPE |
|---|---|---|---|---|---|---|---|
| FastFlow3D* (Jund et al., 2021) | | | Full Point Cloud | 0.071 | 0.186 | 0.021 | 0.006 |
| ZeroFlow 1X* (Ours) | 29.33± | 2.38 | Full Point Cloud | 0.088 | 0.231 | 0.022 | 0.011 |
| ZeroFlow 3X (Ours) | | | Full Point Cloud | 0.064 | 0.164 | 0.017 | 0.011 |
| ZeroFlow 5X (Ours) | | | Full Point Cloud | **0.056** | 0.140 | 0.017 | 0.011 |
| FastFlow3D XL | | | Full Point Cloud | 0.055 | 0.139 | 0.018 | 0.007 |
| ZeroFlow XL 1X (Ours) | 260.61± | 1.21 | Full Point Cloud | 0.070 | 0.178 | 0.019 | 0.013 |
| ZeroFlow XL 3X (Ours) | | | Full Point Cloud | **0.054** | 0.131 | 0.018 | 0.012 |
| NSFP w/ Motion Comp (Li et al., 2021b) | 26,285.0± | 18,139.3 | Full Point Cloud | 0.067 | 0.131 | 0.036 | 0.034 |
| Chodosh et al. (Chodosh et al., 2023) | 35,281.4± | 20,247.7 | Full Point Cloud | 0.055 | 0.129 | 0.028 | 0.008 |
| Odometry | — | | Full Point Cloud | 0.198 | 0.583 | 0.010 | 0.000 |
| ICP (Chen & Medioni, 1992) | 523.11± | 169.34 | Full Point Cloud | 0.204 | 0.557 | 0.025 | 0.028 |
| Gojcic (Gojcic et al., 2021) | 6,087.87± | 1,690.56 | 20000 | 0.083 | 0.155 | 0.064 | 0.032 |
| Sim2Real (Jin et al., 2022) | 99.35± | 13.88 | 8192 | 0.157 | 0.229 | 0.106 | 0.137 |
| EgoFlow (Tishchenko et al., 2020) | 2,116.34± | 292.32 | 8192 | 0.205 | 0.447 | 0.079 | 0.090 |
| PPWC (Wu et al., 2020) | 79.43± | 2.20 | 8192 | 0.130 | 0.168 | 0.092 | 0.129 |
| FlowStep3D (Kittenplon et al., 2021) | 687.54± | 3.13 | 8192 | 0.161 | 0.173 | 0.132 | 0.176 |

As shown in Table 1, ZeroFlow is able to leverage scale to deliver superior performance. While ZeroFlow 1X loses a head-to-head competition against the human-supervised FastFlow3D on both Argoverse 2 (Table 1) and Waymo Open (Table 2), scaling the distillation process to additional unlabeled data provided by Argoverse 2 enables ZeroFlow 3X to significantly surpass the performance of both methods just by training on more pseudo-labled data. ZeroFlow 3X even surpasses the performance of its own teacher, NSFP, *while running in real-time!*

ZeroFlow's pipeline also benefits from scaling up the student architecture. We modify ZeroFlow's architecture with the much larger XL backbone, and show that our ZeroFlow XL 3X is able to

combine the power of dataset and model scale to outperform all other methods, including significantly outperform its own teacher. Our simple approach achieves **state-of-the-art** on both the Argoverse 2 validation split and *Argoverse 2 Self-Supervised Scene Flow Challenge*.

Table 2: Quantitative results on Waymo Open using the evaluation protocol from Chodosh et al. (2023). Runtimes are scaled to approximate the performance on a V100 (Li et al., 2020). Both FastFlow3D and ZeroFlow 1X have identical feedforward architectures and thus share the same runtime. Underlined methods require human supervision.

|  | Runtime (ms) |  | Point Cloud Subsampled Size | Threeway EPE | Dynamic FG EPE | Static FG EPE | Static BG EPE |
|---|---|---|---|---|---|---|---|
| ZeroFlow 1X (Ours) | 21.66± | 0.48 | Full Point Cloud | 0.092 | 0.216 | 0.015 | 0.045 |
| FastFlow3D (Jund et al., 2021) |  |  | Full Point Cloud | 0.078 | 0.195 | 0.015 | 0.024 |
| Chodosh (Chodosh et al., 2023) | 93,752.3± | 76,786.1 | Full Point Cloud | **0.041** | 0.073 | 0.013 | 0.039 |
| NSFP Li et al. (2021b) | 90,999.1± | 74,034.9 | Full Point Cloud | 0.100 | 0.171 | 0.022 | 0.108 |
| ICP (Chen & Medioni, 1992) | 302.70± | 157.61 | Full Point Cloud | 0.192 | 0.498 | 0.022 | 0.055 |
| Gojcic Gojcic et al. (2021) | 501.69± | 54.63 | 20000 | 0.059 | 0.107 | 0.045 | 0.025 |
| EgoFlow (Tishchenko et al., 2020) | 893.68± | 86.55 | 8192 | 0.183 | 0.390 | 0.069 | 0.089 |
| Sim2Real (Jin et al., 2022) | 72.84± | 14.79 | 8192 | 0.166 | 0.198 | 0.099 | 0.201 |
| PPWC (Wu et al., 2020) | 101.43± | 5.48 | 8192 | 0.132 | 0.180 | 0.075 | 0.142 |
| FlowStep3D (Kittenplon et al., 2021) | 872.02± | 6.24 | 8192 | 0.169 | 0.152 | 0.123 | 0.232 |

## 4.2 How does ZeroFlow scale?

Section 4.1 demonstrates that ZeroFlow can leverage scale to capture state-of-the-art performance. However, it's difficult to perform extensive model tuning for large training runs, so predictable estimates of performance as a function of dataset size are critical (OpenAI, 2023). Does ZeroFlow's performance follow predictable scaling laws?

We train ZeroFlow and FastFlow3D on sequence subsets / supersets of the Argoverse 2 Sensor train split. Figure 4 shows ZeroFlow and FastFlow3D's validation Threeway EPE both decrease roughly logarithmically, and this trend appears to hold for XL backbone models as well.

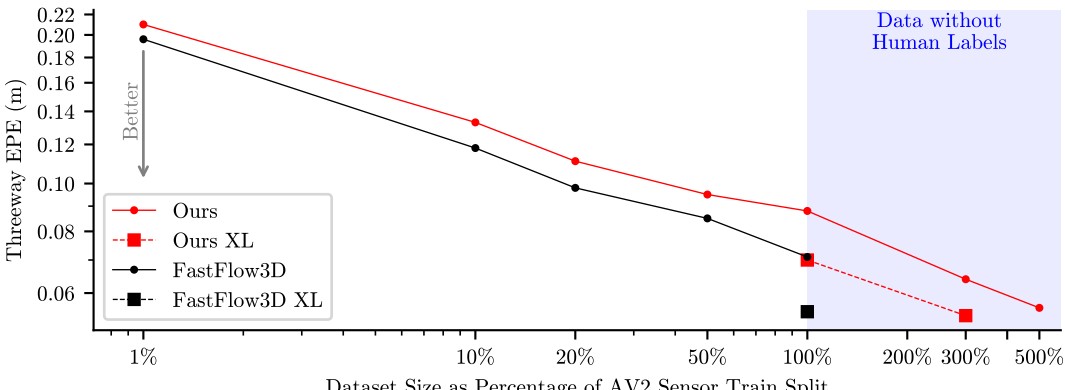

Figure 4: Empirical scaling laws for ZeroFlow. We report Argoverse 2 validation split Threeway EPE as a percentage of the Argoverse 2 *train* split used, on a $\log_{10}$-$\log_{10}$ scale, trained to convergence. Threeway EPE performance of ZeroFlow scales logarithmically with the amount of training data.

Empirically, ZeroFlow adheres to predictable scaling laws that demonstrate more data (and more parameters) are all you need to get better performance. This makes ZeroFlow a practical pipeline for building *scene flow foundation models* (Bommasani et al., 2021) using the raw point cloud data that exists *today* in the deployment logs of Autonomous Vehicles and other deployed systems.

## 4.3 How does dataset diversity influence ZeroFlow's performance?

In typical human annotation setups, a point cloud *sequence* is given to the human annotator. The human generates box annotations in the first frame, and then updates the pose of those boxes as the objects move through the sequence, introducing and removing annotations as needed. This process is much more efficient than annotating disjoint frame pairs, as it amortizes the time spent annotating

most objects in the sequence. This is why most human annotated training datasets (e.g. Argoverse 2 Sensor, Waymo Open) are composed of contiguous *sequences*. However, contiguous frames have significant structural similarity; in the 150 frames (15 seconds) of an Argoverse 2 Sensor sequence, the vehicle typically observes no more than a city block's worth of unique structure. ZeroFlow, which requires *zero* human labels, does not have this constraint on its pseudo-labels; NSFP run on non-sequential frames is no more expensive than NSFP run on non-sequential frames, enabling ZeroFlow to train on a more diverse dataset. How does dataset diversity impact performance?

To understand the impact of data diversity, we train a version of ZeroFlow 1X and ZeroFlow 2X *only* on the diverse subset of our Argoverse 2 LiDAR data selected by uniformly sampling 12 frame pairs from each of the 20,000 unique sequences (Table 3).

Table 3: Comparison between ZeroFlow trained on Argoverse 2 Sensor dataset versus the more diverse, unlabeled Argoverse 2 LiDAR subset described in Section 4.1. Diverse training datasets result in non-trivial performance improvements.

| | Threeway EPE | Dynamic FG EPE | Static FG EPE | Static BG EPE |
|---|---|---|---|---|
| FastFlow3D* (Jund et al., 2021) | 0.071 | 0.186 | 0.021 | 0.006 |
| ZeroFlow 1X (AV2 Sensor Data)* | 0.088 | 0.231 | 0.022 | 0.011 |
| ZeroFlow 1X (AV2 LiDAR Subset Data) | 0.082 | 0.218 | 0.018 | 0.009 |
| ZeroFlow 2X (AV2 LiDAR Subset Data) | 0.072 | 0.184 | 0.022 | 0.011 |

Dataset diversity has a non-trivial impact on performance; ZeroFlow, by virtue of being able to learn across *non-contiguous* frame pairs, is able to see more unique scene structure and thus learn to better to extract motion in the presence of the unique geometries of the real world.

### 4.4 HOW DO THE NOISE CHARACTERISTICS OF ZEROFLOW COMPARE TO OTHER METHODS?

ZeroFlow distills NSFP into a feedforward model from the FastFlow3D family. Section 4.1 highlights the *average* performance of ZeroFlow across Threeway EPE catagories, but what does the error *distribution* look like?

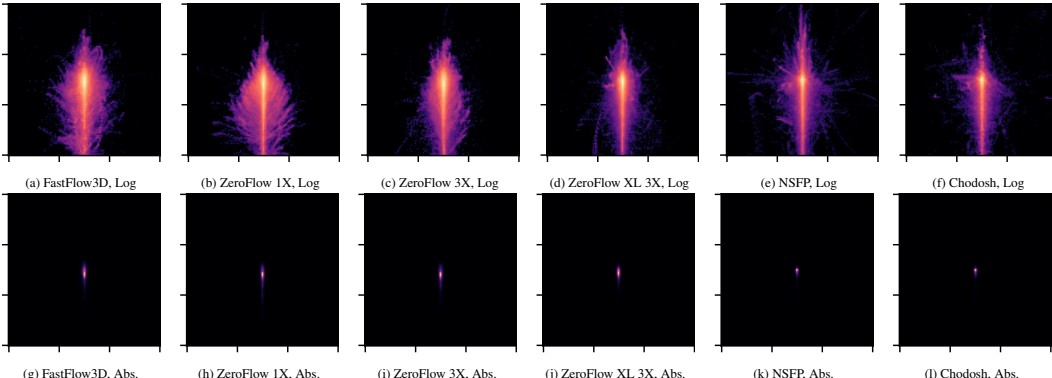

Figure 5: Normalized frame birds-eye-view heatmaps of endpoint residuals for Chamfer Distance, as well as the outputs for NSFP and Chodosh on moving points (points with ground truth speed above 0.5m/s). Perfect predictions would produce a single central dot. Top row shows the frequency on a $\log_{10}$ color scale, bottom row shows the frequency on an absolute color scale. Qualitatively, methods with better quantitative results have tighter residual distributions. See Supplemental F for details.

To answer this question, we plot birds-eye-view flow vector residuals of NSFP, Chodosh, FastFlow3D, and several members of the ZeroFlow family on moving objects from the Argoverse 2 validation dataset, where the ground truth is rotated vertically and centered at the origin to present all vectors in the same frame (Figure 5; see Supplemental F for details on construction). Qualitatively, these plots show that error is mostly distributed along the camera ray and distributional tightness ($\log_{10}$ plots) roughly corresponds to overall method performance.

Overall, these plots provide useful insights to practitioners and researchers, particularly for consumption in downstream tasks; as an example, open world object extraction (Najibi et al., 2022) requires the ability to threshold for motion and cluster motion vectors together to extract the entire object. Decreased average EPE is useful for this task, but understanding the magnitude and *distribution* of flow vectors is needed to craft good extraction heuristics.

### 4.5 HOW DOES TEACHER QUALITY IMPACT ZEROFLOW'S PERFORMANCE?

As shown in Section 4.1 (Chodosh et al., 2023) has superior Threeway EPE over NSFP on both Argoverse 2 and Waymo Open. Can a better performing teacher lead a better version of ZeroFlow?

To understand the impact of a better teacher, we train ZeroFlow on Argoverse 2 using superior quality flow vectors from Chodosh et al. (2023), which proposes a refinement step to NSFP lables to provide improvements to flow vector quality (Table 4). ZeroFlow trained on Chodosh refined pseudo-labels provides no meaningful quality improvement over NSFP pseudo-labels (as discussed in Supplemental C, the Threeway EPE difference is within training variance for ZeroFlow). These results also hold for our ablated speed scaled version of ZeroFlow in Supplemental B.

Since increasing the quality of the teacher over NSFP provides no noticeable benefit, can we get away with using a significantly faster but lower quality teacher to replace NSFP, e.g. the commonly used self-supervised proxy of TruncatedChamfer?

To understand if NSFP is necessary, we train ZeroFlow on Argoverse 2 using pseudo-labels from the nearest neighbor, truncated to 2 meters as with TruncatedChamfer. The residual distribution of TruncatedChamfer is shown in Supplemental F, Figure 10a. ZeroFlow trained on TruncatedChamfer pseudo-labels performs significantly worse than NSFP, motivating the use of NSFP as a teacher.

Table 4: Comparison between ZeroFlow trained on Argoverse 2 using NSFP pseudo-labels, ZeroFlow using Chodosh et al. (2023) pseudo-labels, and ZeroFlow using TruncatedChamfer. Methods with an * have performance averaged over 3 training runs (see Supplemental C for details). The minor quality improvement of Chodosh pseudo-labels does not lead to a meaningful difference in performance, while the significant degradation of TruncatedChamfer leads to significantly worse performance.

| | Threeway EPE | Dynamic FG EPE | Static FG EPE | Static BG EPE |
|---|---|---|---|---|
| ZeroFlow 1X (NSFP pseudo-labels)* | 0.088 | 0.231 | 0.022 | 0.011 |
| ZeroFlow 1X (Chodosh et al. (2023) pseudo-labels) | 0.085 | 0.234 | 0.018 | 0.004 |
| ZeroFlow 1X (TruncatedChamfer pseudo-labels) | 0.105 | 0.226 | 0.049 | 0.040 |

## 5 CONCLUSION

Our scene flow approach, Zero-Label Scalable Scene Flow (ZeroFlow), produces fast, state-of-the-art scene flow *without human labels* via our conceptually simple distillation pipeline.

But, more importantly, we present the first practical pipeline for building *scene flow foundation models* (Bommasani et al., 2021) using the raw point cloud data that exists *today* in the deployment logs at Autonomous Vehicle companies and other deployed robotics systems. Foundational models in other domains like language (Brown et al., 2020; OpenAI, 2023) and vision (Kirillov et al., 2023; Rajeswaran et al., 2022) have enabled significant system capabilities with little or no additional domain-specific fine-tuning (Wang et al., 2023; Ma et al., 2022; 2023). We posit that a scene flow foundational model will enable new systems that can leverage high quality, general scene flow estimates to robustly reason about object dynamics even in foreign or noisy environments.

**Limitations and Future Work.** ZeroFlow inherits the biases of its pseudo-labels. Unsurprisingly, if the pseudo-labels consistently fail to estimate scene flow for certian objects, our method will also be unable to predict scene flow for those objects; however, further innovation in model architecture, loss functions, and pseudo-labels may yield better performance. In order to enable further work on Scene Flow via Distillation-based methods, we release[3] our code, trained model weights, and NSFP flow pseudo-labels, representing 3.6 GPU months for Argoverse 2 and 3.5 GPU months for Waymo Open.

---

[3] https://vedder.io/zeroflow

**Acknowledgements.** The research presented in this paper was partially supported by the DARPA SAIL-ON program under contract HR001120C0040, the DARPA ShELL program under agreement HR00112190133, the Army Research Office under MURI grant W911NF20-1-0080, and the CMU Center for Autonomous Vehicle Research. This work was performed under the auspices of the U.S. Department of Energy by Lawrence Livermore National Laboratory under Contract DE-AC52-07NA27344.

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
