## A    ARGOVERSE 2 AND WAYMO OPEN DATASET CONFIGURATION DETAILS

**Argoverse 2.** The Sensor dataset contains 700 training and 150 validation sequences. Each sequence contains 15 seconds of 10Hz point clouds collected using two Velodyne VLP-32s mounted on the roof of a car. As part of the training protocol for ZeroFlow, FastFlow3D, and NSFP w/ Motion Compensation, we perform ego compensation, ground point removal, and restrict all points to be within a 102.4m × 102.4m area centered around the ego vehicle, resulting in point clouds with an average of 52,871 points (Figure 6a). The point cloud $P_{t+1}$ is centered at the origin of the ego vehicle's coordinate system and $P_t$ is projected into $P_{t+1}$'s coordinate frame. For ZeroFlow and FastFlow3D, the PointPillars encoder uses 0.2m×0.2m pillars, with all architectural configurations matching (Jund et al., 2021). For NSFP w/ Motion Compensation, we use the same architecture and early stopping parameters as the original method (Li et al., 2021b). For FastFlow3D and the FastFlow3D student architecture of ZeroFlow, we train to convergence (50 epochs) with an Adam (Kingma & Ba, 2014) learning rate of $2 \times 10^{-6}$ and batch size 64. For FastFlow3D XL and the FastFlow3D XL student architecture of ZeroFlow (ZeroFlow XL 1X, ZeroFlow XL 3X), we train to convergence (10 epochs) with the same optimizer settings and a batch size 12. For ZeroFlow 3X and and ZeroFlow XL 3X, we train on an additional 240,000 unlabeled frame pairs (roughly twice the size as the Argoverse 2 Sensor *train* split), constructed by selecting 12 frame pairs at uniform intervals from the 20,000 sequences of the Argoverse 2 LiDAR dataset. For all other methods in Table 1, we use the implementations provided by Chodosh et al. (2023), which follow ground removal and ego compensation protocols from their respective papers.

**Waymo Open.** The dataset contains 798 training and 202 validation sequences. Each sequence contains 20 seconds of 10Hz point clouds collected using a custom LiDAR mounted on the roof of a car. We use the same preprocessing and training configurations used on Argoverse 2; after ego motion compensation and ground point removal, the average point cloud has 79,327 points (Figure 6b).

As shown in Figure 6, Argoverse 2 (Wilson et al., 2021) and Waymo Open (Sun et al., 2020) are significantly larger than the 8,192 point subsampled point clouds used by prior art.

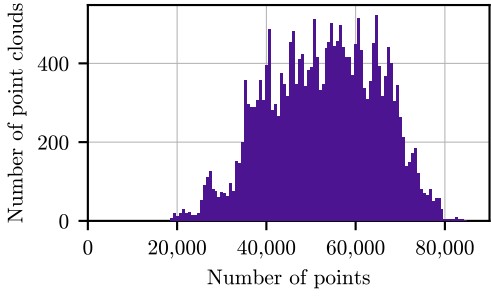 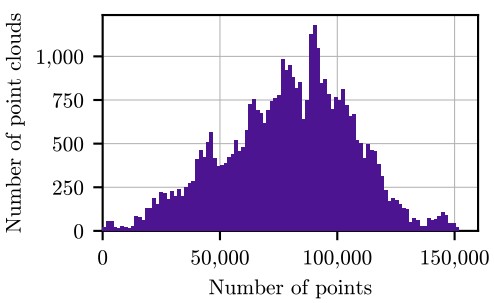

(a) Distribution of point cloud sizes in the Argoverse 2 Sensor *val* split: $\mu = 52,871.6; \sigma = 12,227.2$.

(b) Distribution of point cloud sizes in the Waymo Open *val* split: $\mu = 79,327.8; \sigma = 27,182.1$.

Figure 6: Point cloud size distributions for the *val* set of the Argoverse 2 Sensor (Wilson et al., 2021) and Waymo Open (Sun et al., 2020) datasets after ground removal and clipped to a 102.4m × 102.4m box around the ego vehicle.

## B    EXPLORING THE IMPORTANCE OF POINT WEIGHTING

In order to train FastFlow3D using pseudo-labels, we need a replacement $\sigma(\cdot)$ semantics scaling function described in Equation 4) because our pseudo-labels do not provide foreground / background semantics. In the main experiments, we use uniform scaling ($\sigma(\cdot) = 1$).

### B.1    CAN WE DESIGN A BETTER POINT WEIGHTING FUNCTION FOR PSEUDO-LABELS?

We propose a soft weighting based on pseudo-label flow magnitude: for the point $p$ in the pseudo-label flow $F^*_{t,t+1}(p)$, where $s(p)$ represents its speed in meters per second, we linearly interpolate the

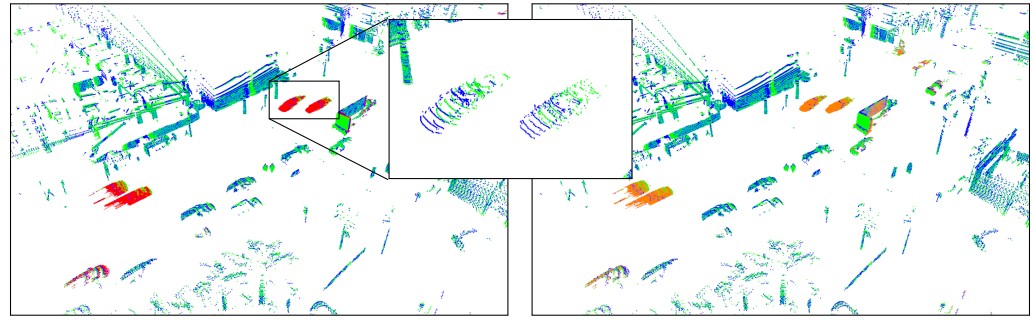

Figure 7: Scene flow estimation of two consecutive point clouds sampled 100 ms apart (green and blue, respectively) on Argoverse 2 (Wilson et al., 2021). **Left:** Ground truth scene flow annotations in red. These annotations are derived from the motion of amodal bounding boxes. **Right:** ZeroFlow's scene flow estimates estimates in orange, which closely match with the ground truth.

weight of $p$ between $0.1\times$ at 0.4 m/s and full weight at 1.0 m/s, i.e.

$$\sigma(p) = \begin{cases} 0.1 & \text{if } s(p) < 0.4 \text{ m/s} \\ 1.0 & \text{if } s(p) > 1.0 \text{ m/s} \\ 1.8s - 0.8 & \text{o.w.} \end{cases} \quad (6)$$

These thresholds are selected to down-weight approximately 80% of points by $0.1\times$, with the other 20% of points split between the soft and full weight region[4]. In Table 5, we show that our weighting scheme provides non-trivial improvements over uniform weighting (i.e. $\sigma(\cdot) = 1$) for ZeroFlow 1X; however, it actually hurts performance for ZeroFlow 3X.

Table 5: Comparison between ZeroFlow trained on Argoverse 2 using NSFP pseudo-labels and ZeroFlow using Chodosh et al. (2023) pseudo-labels using both uniform and speed scaled point weighting. Methods with an * have performance averaged over 3 training runs (see Supplemental C for details).

| | Threeway EPE | Dynamic FG EPE | Static FG EPE | Static BG EPE |
|---|---|---|---|---|
| ZeroFlow 1X (Equation 6, NSFP pseudo-labels)* | 0.084 | 0.217 | 0.023 | 0.011 |
| ZeroFlow 1X (Equation 6, Chodosh et al. (2023) pseudo-labels) | 0.086 | 0.227 | 0.019 | 0.011 |
| ZeroFlow 1X (NSFP pseudo-labels)* | 0.088 | 0.231 | 0.022 | 0.011 |
| ZeroFlow 1X (Chodosh et al. (2023) pseudo-labels) | 0.085 | 0.234 | 0.018 | 0.004 |
| ZeroFlow XL 3X | 0.053 | 0.131 | 0.018 | 0.011 |
| ZeroFlow XL 3X (Equation 6) | 0.056 | 0.139 | 0.017 | 0.011 |

## B.2 HOW MUCH OF FASTFLOW3D'S PERFORMANCE IS DUE TO ITS SEMANTIC POINT WEIGHTING?

Unlike ZeroFlow, FastFlow3D *can* use human foreground / background point labels to upweight the flow importance of foreground points (Section 3.3, Equation 4). To understand the impact of this weighting, we train FastFlow3D with two modified losses; rather than scaling using semantics as described in Equation 4, we uniformly weight all points ($\sigma(\cdot) = 1$) or our speed based weighting (Equation 6).

As shown in Table 6, the performance of FastFlow3D ($\sigma(\cdot) = 1$) and (Equation 6) degrades more than halfway to ZeroFlow's performance.

This raises the question: why is the performance improvement of semantic weighting larger than the improvement of our unsupervised moving point weighting scheme (Supplemental B.1)? We

---

[4]For Argoverse 2, exactly 78.1% of points are downweighted, 11.8% lie in the soft-weight region, and 10.1% lie in the full weight region; for Waymo Open 80.0% of points are downweighted, 7.9% lie in the soft-weight region, and 12.1% lie in the full-weight region respectively.

Table 6: Comparison between ZeroFlow, FastFlow3D, and the ablated FastFlow3D with uniform scaling ($\sigma(\cdot) = 1$) trained on Argoverse 2. The performance of FastFlow3D with Uniform Scaling and our speed scaling (Equation 6) are nearly identical to ZeroFlow's performance. Methods with an * have performance averaged over 3 training runs (see Supplemental C for details). Underlined methods require human supervision.

|  | Threeway EPE | Dynamic FG EPE | Static FG EPE | Static BG EPE |
| --- | --- | --- | --- | --- |
| ZeroFlow 1X* (Ours) | 0.088 | 0.231 | 0.022 | 0.011 |
| FastFlow3D ($\sigma(\cdot) = 1$) | 0.081 | 0.220 | 0.018 | 0.006 |
| FastFlow3D (Equation 6) | 0.081 | 0.224 | 0.018 | 0.002 |
| FastFlow3D* (Jund et al., 2021) | 0.071 | 0.186 | 0.021 | 0.006 |

conjecture that not only does semantic weighting provide increased loss on moving objects, it implicitly teaches the network to recognize the structure of objects themselves. For example, with Equation 4 scaling, end-point error on a stationary pedestrian is significantly higher than static background points, incentivizing the network to learn to detect the point *structure* common to pedestrians, even if immobile, to perfect the predictions on those points.

## C  CHARACTERIZING INTER-TRAINING RUN FINAL PERFORMANCE VARIANCE FOR ZEROFLOW AND FASTFLOW3D

On Argoverse 2, Threeway EPE difference between ZeroFlow and the human supervised FastFlow3D is 1.6cm (Table 1); how much of this gap can be attributed to training variance between runs? To answer this question, we train ZeroFlow and FastFlow3D from scratch 3 times each. ZeroFlow is trained on the same Argoverse 2 NSFP pseudo-labels (Table 7), resulting in a mean Threeway EPE of 0.088m with error of 0.003m (0.3cm) in either direction, and FastFlow3D is trained on the Argoverse 2 human labels (Table 9), resulting in a mean Threeway EPE of 0.071m with error under 0.003m (0.3cm) in either direction.

To contextualize the scale of this variance, the underlying Velodyne VLP-32 sensors used to collect the Argoverse 2 are only certified to ±3 cm of error (Lopac et al., 2022) (an order of magnitude greater than the deviation from the mean train performance for ZeroFlow), and this entirely neglects additional sources of noise introduced from other real world effects such as empirical ego motion compensation.

Table 7: Performance of ZeroFlow over 3 train runs on the same NSFP pseudo-labels.

|  | Threeway EPE | Dynamic FG EPE | Static FG EPE | Static BG EPE |
| --- | --- | --- | --- | --- |
| ZeroFlow 1X Run #1 | 0.085 | 0.224 | 0.021 | 0.011 |
| ZeroFlow 1X Run #2 | 0.088 | 0.231 | 0.022 | 0.010 |
| ZeroFlow 1X Run #3 | 0.091 | 0.240 | 0.023 | 0.011 |
| ZeroFlow 1X Average | 0.088 | 0.231 | 0.022 | 0.011 |

Table 8: Performance of ZeroFlow ablated with point scaling (Equation 6) over 3 train runs on the same NSFP pseudo-labels.

|  | Threeway EPE | Dynamic FG EPE | Static FG EPE | Static BG EPE |
| --- | --- | --- | --- | --- |
| ZeroFlow 1X (Equation 6) Run #1 | 0.083 | 0.214 | 0.023 | 0.011 |
| ZeroFlow 1X (Equation 6) Run #2 | 0.083 | 0.215 | 0.024 | 0.011 |
| ZeroFlow 1X (Equation 6) Run #3 | 0.085 | 0.222 | 0.022 | 0.011 |
| ZeroFlow 1X (Equation 6) Average | 0.084 | 0.217 | 0.023 | 0.011 |

Table 9: Performance of FastFlow3D over 3 train runs on the Argoverse 2 human labels.

|  | Threeway EPE | Dynamic FG EPE | Static FG EPE | Static BG EPE |
|---|---|---|---|---|
| FastFlow3D Run #1 | 0.070 | 0.181 | 0.020 | 0.006 |
| FastFlow3D Run #2 | 0.071 | 0.186 | 0.021 | 0.007 |
| FastFlow3D Run #3 | 0.073 | 0.191 | 0.023 | 0.006 |
| FastFlow3D Average | 0.071 | 0.186 | 0.021 | 0.006 |

# D   CHARACTERIZING HOW ZEROFLOW'S PERFORMANCE EVOLVES DURING TRAINING

Threeway EPE breaks down performance into three categories: *Foreground Dynamic*, *Foreground Static*, and *Background*. How does ZeroFlow's performance evolve during training?

To understand this, we plot ZeroFlow 1X and ZeroFlow 3X in Figure 8. Both methods converge to their final background performance almost immediately, and most of the improvements seen in the final Threeway EPE stem from improvements in Foreground Dynamic (Figure 8b). The impact of additional data is also made clear early in training, as ZeroFlow 3X has significantly lower Threeway EPE by epoch 15 than ZeroFlow 1X.

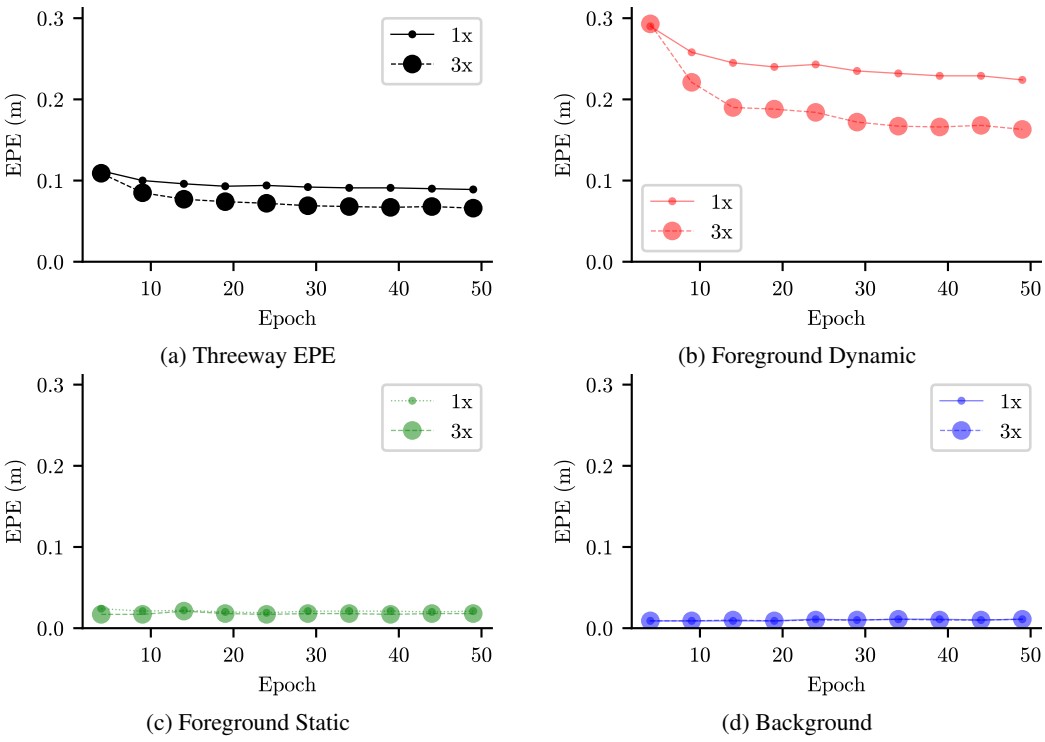

Figure 8: Performance of ZeroFlow 1X and ZeroFlow 3X on the Argoverse 2 *val* split by training epoch. Both methods converge to their final background performance almost immediately, and most of the improvements seen in the final Threeway EPE stem from improvements in Foreground Dynamic (Figure 8b).

# E   ESTIMATING HUMAN LABELING VERSUS PSEUDO-LABELING COSTS

NSFP pseudolabeling of the Argoverse 2 train split (700 sequences of 150 frames) required a total of 753 hours of NVidia Turing generation GPU time. At September, 2023 Amazon Web Services EC2

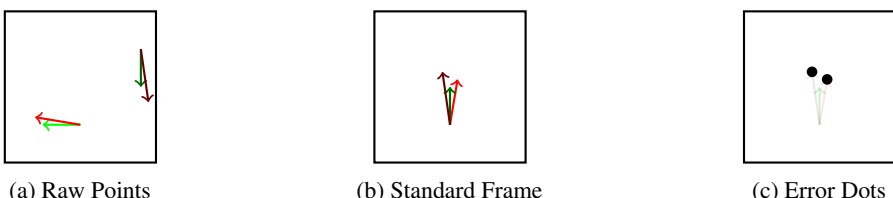

(a) Raw Points         (b) Standard Frame         (c) Error Dots

Figure 9: Process for constructing the endpoint residual plots. The raw points (Figure 9a) are transformed into a standard frame with the ground truth vector pointing up and the endpoint at the center of the plot (Figure 9b), and the residual endpoints are accumulated (Figure 9c).

prices, a single `g4dn.xlarge`, equipped with a single NVidia Tesla T4, costs $0.526 per hour[5], for a total cost of $394 to pseudo-label. By comparison, at an estimated $0.10 per frame per cuboid (no public cost statements exist for production quality AV dataset labels, but this the standard price point within the industry), Argoverse 2's train split has an average of 75 cuboids per frame (Wilson et al., 2021), for a total cost on the order of $787,500 to human annotate.

## F DETAILS ON ENDPOINT RESIDUALS

The process of constructing these endpoint residual plots is shown in Figure 9. For moving points (points with a ground truth flow vector magnitude >0.5m/s), the raw points (Figure 9a) are transformed into a standard frame with the ground truth vector pointing up and the endpoint at the center of the plot (Figure 9b), and the residual endpoints are accumulated (Figure 9c). Residual plots for baselines, as well as their unrotated counterparts, are shown in Figure 10.

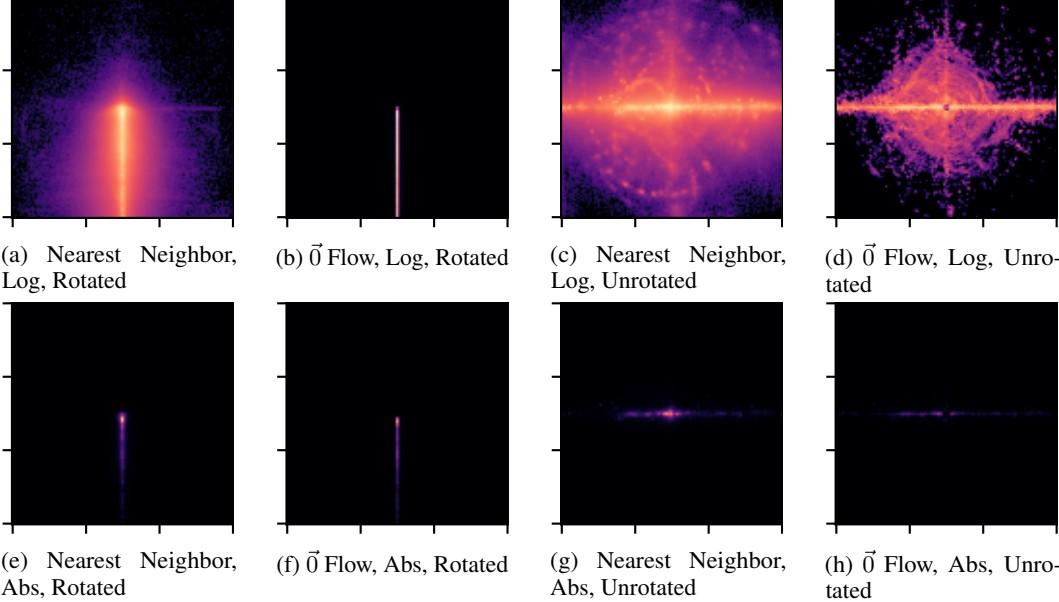

(a) Nearest Neighbor, Log, Rotated

(b) $\vec{0}$ Flow, Log, Rotated

(c) Nearest Neighbor, Log, Unrotated

(d) $\vec{0}$ Flow, Log, Unrotated

(e) Nearest Neighbor, Abs, Rotated

(f) $\vec{0}$ Flow, Abs, Rotated

(g) Nearest Neighbor, Abs, Unrotated

(h) $\vec{0}$ Flow, Abs, Unrotated

Figure 10: Birds-eye-view heatmap of endpoint residuals for naïve flow methods of predicting flow (Nearest Neighbor and $\vec{0}$ Flow on all points) for non-background points moving above 0.5m/s in the raw coordinate frame of the ground truth labels. Brighter color indicates more points in each bin. Perfect labels would produce a single central dot. Distance between ticks is 1 meter. Top row shows frequency on a log color scale to display error distribution shape. Bottom row shows frequency on an absolute color scale to display centroid. Left half shows results in the rotated ground truth coordinate frame. Right half shows results in the unrotated ground truth coordinate frame.

---

[5]https://aws.amazon.com/ec2/pricing/on-demand/

# G  FAQ

## G.1  Our method is "just" a combination of existing methods using standard distillation. Where does the novelty come in?

Michael Black argues that "the simplicity of an idea is often confused with a lack of novelty when exactly the opposite is often true." (Black, 2022). Indeed, we think our novelty comes from the fact that our simple and post-hoc obvious pipeline produces surprisingly good results; our simple pipeline need only consume more raw data to improve and capture state-of-the-art over expensive human supervision while using the same feedforward model architectures.

## G.2  What are the fundamental insights from this paper? What new knowledge was generated?

Beyond producing a useful artifact, our straight-forward pipeline shows that simply training a supervised model with imperfect pseudo-labels can *exceed* the performance of perfect human labels on substantial fraction of the data. We think this is itself surprising, but we also think it has highly impactful implications for the problem of scene flow estimation: *point cloud quantity and diversity is more important than perfect flow label quality for training feedforward scene flow estimators*.

We also think this statement and our empirical scaling laws (Section 4.2) lead directly to actionable advice for practitioners at Autonomous Vehicle companies and other organizations with a large trove of diverse point cloud data: *scaling ZeroFlow on this large scale data will net a significantly better scene flow estimator than expensive human supervision would using a 1000× larger budget*.

In addition to insights, we also present a novel scene flow estimation analysis technique. To our knowledge, the residual plots in Section 4.4 are the first attempt at visualizing the residual *distribution* of scene flow estimators. We think these plots provide useful insights to practitioners and researchers, particularly for consumption in downstream tasks; as an example, open world object extraction (Najibi et al., 2022) requires the ability to threshold for motion and cluster motion vectors together to extract the entire object. Decreased average EPE is useful for this task, but understanding the *distribution* of flow vectors is needed to craft good extraction heuristics.