# OpenReview forum: "ZeroFlow: Scalable Scene Flow via Distillation"
_ICLR.cc/2024/Conference — ICLR 2024 poster_

### Official Review · Reviewer_pGhB · 2023-10-28

**Soundness:** 3 good
**Presentation:** 3 good
**Contribution:** 1 poor
**Rating:** 5
**Confidence:** 2

**Summary:**

This paper focuses on tackling the practical challenge of real-world scale scene flow. The method employed in this study can be summarized as follows: 1) a slow teacher NSFP is identified; 2) a fast student FastFlow3D is recognized; 3) the teacher is adopted to create auto-labels for supervising the student.

**Strengths:**

This paper carries out a thorough series of experiments to validate the efficacy of the straightforward autolabel concepts and raises several intriguing questions. Furthermore, the paper includes experiments designed to address these questions, potentially offering valuable insights to the autonomous vehicle industry.

**Weaknesses:**

The idea lacks novelty, particularly for consideration at ICLR. In my opinion, this paper would be better suited for ICRA or WACV, as they tend to focus on more application and system-oriented research.

**Questions:**

1. For the full point cloud setting, what is the exact number of points in average? Have we applied ground point removal so the number of points has been significantly reduced. I know there is a discussion in 3.1, but I want to confirm the actual number of points and settings in the later experiments.
2. Algorithm 1 can be removed; it does not provide any useful information.
3. (This is not important) I feel we should not call this scene flow foundation model; scene flow is only a single problem in Computer Vision/Perception.

---

> ### Author Response · Authors · 2023-11-13
>
> Thank you for your review.
>
> # Idea lacks novelty
>
> Michael Black’s reviewer guidelines [1] eloquently argue that "the simplicity of an idea is often confused with a lack of novelty when exactly the opposite is often true. Such reviews lead researchers to make an idea appear more complex so that reviewers will find it of higher value."
>
> Indeed, our novelty comes from the fact that our simple and post-hoc obvious pipeline produces state-of-the-art results. To quote q16v: "This distillation idea is very simple, but previous work on point-cloud-based scene flow methods hasn't tried the idea yet. The paper demonstrates that this simple idea is working and demonstrates that it's scalable by using large-scale diverse unlabeled data." This result also produces "valuable insights to the autonomous vehicle industry" (pGhB).
>
>
> #  This paper would be better suited for ICRA or WACV, as they tend to focus on more application and system-oriented research
>
> Our paper provides non-trivial, fundamental insights: our straight-forward pipeline shows that simply training a supervised model with imperfect pseudo-labels can *exceed* the performance of perfect human labels given sufficient unlabeled data. We think this is itself surprising, but we also think it has significant implications for the problem of scene flow estimation in general: *point cloud quantity and diversity is more important than perfect flow label quality for training feedforward scene flow estimators*. We translate this into an empirical scaling law that describes the log-log relationship between dataset size and final Threeway EPE performance.
>
> We also present a novel scene flow estimation analysis technique. To our knowledge, the residual plots in Section 4.4 are the first attempt at visualizing the residual *distribution* of scene flow estimators. We think these plots provide useful insights to practitioners and researchers. For example, open world object extraction [2] must be able to threshold and  cluster motion vectors together to extract the entire object. Understanding the *distribution of flow vectors* is essential for identifying extraction heuristics. .
>
> Furthermore, we provide extensive scientific inquiries on the impact of not only method component choices (e.g. Teacher quality — Section 4.5), but of the data itself (e.g. Dataset diversity — Section 4.3). These inquiries provide important insights to practitioners and researchers on what are fruitful directions to explore.
>
> We believe this makes our paper of interest to the ICLR community looking at the "applications to robotics, autonomy, planning" primary area.
> # Numbers on dataset point cloud sizes
>
> We provide plots of the full point cloud size distribution for AV2 and Waymo datasets after ground point removal and region clipping (102.4m x 102.4m area around the ego vehicle) in the Supplemental, Section A. The average number of points per point cloud is 52,871.6 points for AV2 and 79,327.8 points for Waymo Open.
>
> # Removing Algorithm 1
>
> Thank you for pointing this out. We will address this for the camera ready.
>
> [1] Michael Black. Novelty in science: A guide to reviewers.

---

> ### Comment · Area_Chair_iY4a · 2023-11-21
>
> Reviewer pGhB, did the authors' rebuttal had addressed your concerns and questions?
>
> Please reply and post your final decision.
>
> AC

---

### Official Review · Reviewer_E12X · 2023-10-30

**Soundness:** 2 fair
**Presentation:** 3 good
**Contribution:** 2 fair
**Rating:** 5
**Confidence:** 4

**Summary:**

The paper proposes a new method called ZeroFlow that combines the strengths of optimization-based and feed-forward methods to achieve efficient and label-free scene flow estimation.
It introduces an optimization-based teacher method, which generates pseudo labels, and uses these labels to supervise a feed-forward student model.
It conducts extensive quantitative evaluations and achieves comparable performance to NSFP and FastFlow3D.

**Strengths:**

- The overall writing of the paper is good, with a clear description of the motivation and implementation of the proposed approach.
- As can be seen from Table 1, the proposed method achieves a good balance between performance and efficiency.
- The paper verifies the proposed ideas through a series of experiments.

**Weaknesses:**

- The innovation and contribution of the paper mainly lie in the new application of the distillation strategy, and I suggest the authors provide further analysis or insights into the scene flow estimation task.
I think the achieved performance of the proposed approach is highly dependent on the quality and effectiveness of the existing works.
The authors combine two previously published works, using NSFP to generate pseudo labels and training the FastFlow3D model using these labels.

- In Sections 3.2 and 3.3, the author provides a detailed description to help understand the proposed method, but it is difficult for me to detect the author's new in-depth thinking about this work from these descriptions.
At the same time, Section 3 contains many statements on existing work, such as 3.1. This section might benefit from providing a more in-depth analysis of combining the strengths of optimization-based and feed-forward methods.

- The quantitative results in Table 2 on Waymo Open are less competitive, making the method's performance on this dataset not very convincing.

- The overall style of the article is more like a technical report than a research paper, and I feel that it contains more engineering skills.

**Questions:**

Please refer to the Weaknesses

---

> ### Author Response · Authors · 2023-11-13
>
> Thank you for your review.
> # Novelty
>
> Michael Black’s reviewer guidelines [1] eloquently argue that "the simplicity of an idea is often confused with a lack of novelty when exactly the opposite is often true. Such reviews lead researchers to make an idea appear more complex so that reviewers will find it of higher value."
>
> Indeed, our novelty comes from the fact that our simple and post-hoc obvious pipeline produces state-of-the-art results. To quote q16v: "This distillation idea is very simple, but previous work on point-cloud-based scene flow methods hasn't tried the idea yet. The paper demonstrates that this simple idea is working and demonstrates that it's scalable by using large-scale diverse unlabeled data." This result also produces "valuable insights to the autonomous vehicle industry" (pGhB).
>
> # Performance of our system is the result of its components
>
> Our performance cannot be strictly attributed to the components plugged into our *Scene Flow via Distillation* framework — we benchmark ZeroFlow and outperform both NSPF and FastFlow3D.
>
> But, broadly, we agree that we benefit from high quality components! Indeed, a feature of our *Scene Flow via Distillation* pipeline is, given a better test-time optimization method, a better supervised scene flow network architecture, or more unlabeled data, it can leverage these component improvements to produce an even better final method *for free*.
> # More analysis of of the combined strengths of optimization-based and feed-forward methods
> We refer the reviewer to Section 3.1, which analyzes the combined strengths of optimization-based and feed forward methods. Optimization methods produce high quality flow, but have high computational cost, whereas feed-forward methods are fast but limited by the scale of their supervised labels. Combining these two techniques gives us a formula for scaling that allows our combined method to exceed the performance of its components and achieve state-of-the-art results. We would gladly appreciate suggestions for further analysis.
> # Waymo Open (Table 2) results not competitive
>
> ZeroFlow provides a label-free, scalable training pipeline. We expect that, relative to human supervision *on the same data*, ZeroFlow's pseudolabeling approach will perform worse.
>
> This is reflected in ZeroFlow's performance on Argoverse 2 (Table 1); ZeroFlow 1x performs worse than its supervised counterpart, FastFlow3D. However, when scaling up training to include the unannotated Argoverse 2 LiDAR dataset, ZeroFlow exhibits meaningfully superior performance compared to the human supervised counterpart (ZeroFlow 3X vs FastFlow3D).
>
> This same pattern is also reflected in Table 2 for Waymo Open. Waymo Open does not provide any additional unlabeled data beyond their annotated dataset, preventing us from scaling up ZeroFlow further; however, our performance results on the data we do have are actually quite positive. Supervised FastFlow3D has a very similar Threeway EPE on both datasets, while ZeroFlow 1x on Waymo Open is actually closer in performance to FastFlow3D than on AV2 where we successfully demonstrated that more raw data produces state-of-the-art performance.
>
> # This is a technical report not a research paper; mostly engineering
>
> Our paper provides non-trivial, fundamental insights: our straight-forward pipeline shows that simply training a supervised model with imperfect pseudo-labels can *exceed* the performance of perfect human labels given sufficient unlabeled data. We think this is itself surprising, but we also think it has significant implications for the problem of scene flow estimation in general: *point cloud quantity and diversity is more important than perfect flow label quality for training feedforward scene flow estimators*. We translate this into an empirical scaling law that describes the log-log relationship between dataset size and final Threeway EPE performance.
>
> We also present a novel scene flow estimation analysis technique. To our knowledge, the residual plots in Section 4.4 are the first attempt at visualizing the residual *distribution* of scene flow estimators. We think these plots provide useful insights to practitioners and researchers. For example, open world object extraction [2] must be able to threshold and  cluster motion vectors together to extract the entire object. Understanding the *distribution of flow vectors* is essential for identifying extraction heuristics. .
>
> Furthermore, we provide extensive scientific inquiries on the impact of not only method component choices (e.g. Teacher quality — Section 4.5), but of the data itself (e.g. Dataset diversity — Section 4.3). These inquiries provide important insights to practitioners and researchers on what are fruitful directions to explore.
>
> [1] Michael Black. Novelty in science: A guide to reviewers.
> [2] Najibi et al. Motion Inspired Unsupervised Perception and Prediction in Autonomous Driving. European Conference on Computer Vision. ECCV 2022.

---

> ### Comment · Area_Chair_iY4a · 2023-11-21
>
> Reviewer E12X, did the authors' rebuttal had addressed your concerns?
> Please reply to the author and post the final decision.
>
> AC

---

### Official Review · Reviewer_q16v · 2023-11-01

**Soundness:** 3 good
**Presentation:** 3 good
**Contribution:** 3 good
**Rating:** 8
**Confidence:** 4

**Summary:**

The paper introduces a scalable point-cloud-based scene flow approach via distillation. Given unlabeled raw point cloud data, the paper first calculates pseudo-GT scene flow using a test-time optimization-based method (Neural Scene Flow Prior), which tends to demonstrate much better accuracy than feed-forward approaches. Then given the pseudo GT, the method trains a feed-forward model in a supervised manner. With this distillation pipeline, their method archives comparable/sometimes even better accuracy than the optimization-based method and of course much better than other feed-forward methods, while maintaining real-time performance. This pipeline is scalable as what it needs is just unlabeled point cloud data.

**Strengths:**

+ Good performance (Table 1)

   The method achieves better accuracy over previous work while maintaining real-time performance. This source of gain requires the cost and time of generating pseudo-GT using the existing optimization-based approach. However, this can be done offline in a parallel way, so it's not a critical concern.

+ Simple, effective idea

   This distillation idea is very simple, but previous work on point-cloud-based scene flow methods hasn't tried the idea yet. The paper demonstrates that this simple idea is working and demonstrates that it's scalable by using large-scale diverse unlabeled data. It would be great if the computed pseudo-GT would be released to the community.

+ In-depth analysis

  Not only the main experiments, the paper provides in-depth analysis, such as accuracy versus dataset size (Fig. 4), endpoint residual maps (Fig. 5), and variance study (Table 7 and 8). This clearly helps a better understanding of the paper. The paper is pretty much self-contained.

**Weaknesses:**

There seem to be not many weaknesses in the paper but some minor questions related to the accuracy.

- In Table 1, how can ZeroFlow XL 3X (Ours) outperform the NSFP w/ Motion Comp baseline as ZeroFlow's accuracy can be bounded by the accuracy of pseudo GT (i.e., NSFP)? Would it be also possible to include the accuracy of pseudo GT in Table 1?

- In Fig. 4, why is FastFlow3D better than 'Ours' when the dataset size is less than 100%? I thought ZeroFlow's backbone was based on FastFlow3D, but what change made it underperform FastFlow3D?

**Questions:**

(just a remark) By the way, there is a recent paper [a] that significantly improves the runtime performance of NSFP by 10 or 30 times (varying different datasets). The proposed approach can reduce the compute cost for pseudo GT generation by using this faster method.

[a] Fast Neural Scene Flow, ICCV 2023

---

> ### Author Response · Authors · 2023-11-13
>
> Thank you for your review. As mentioned in the abstract, we intend to release all of our source code along with the pseudolabels we used to train ZeroFlow.
> # Why does ZeroFlow's scaled up performance surpass the teacher's performance?
>
> ZeroFlow's student-teacher distillation setup enables the student to surpass the teacher because neural networks are good at learning from noisy labels [1]. As we show in Figure 5, as ZeroFlow's pseudolabeled dataset is scaled up (1x, 3x), the variance of its error distribution shrinks because the student gets better at predicting the *expected value* of the NSFP motion estimates conditioned on the observation of motion. This lower variance expected value estimation leads to lower Threeway EPE estimates compared to even NSFP itself.
> # Pseudolabel performance in Table 1
>
>  As shown in Table 1, NSFP (our pseudolabeler) achieves 0.068 EPE.
> # Why does FastFlow3D outperform ZeroFlow when the dataset size is less than 100%?
>
> FastFlow3D is trained using ground truth flow provided via human annotations. ZeroFlow is trained using pseudolabel flow provided via NSFP. NSFP is an imperfect pseudolabeler, so even though ZeroFlow and FastFlow3D share the same network architecture and see the same point clouds, ZeroFlow's supervision is from inferior quality pseudolabels. Therefore,  performance is expected to be worse than its human supervised counterpart.
>
> # Faster teacher method
>
> Thank you for this reference! We will look into this as a faster teacher!
>
> [1] Li et al. Learning from Noisy Labels with Distillation. ICCV 2017.

---

> ### Comment · Area_Chair_iY4a · 2023-11-21
>
> Reviewer q16v, while the author had posted the rebuttal, please reply if you still have any concerns, or post your final decision based on the reviews, rebuttals and the revision.
>
> AC

---

> ### Comment · Reviewer_q16v · 2023-11-22
> **Response**
>
> The comments by the authors fully resolve my main concerns on technical aspects as well as on results. I keep my original rating, Accept.
>
> I also read other reviews and responses to them; I understand the concerns of other reviewers, but I think the responses are reasonable.
>
> In terms of technical aspects, I agree that the paper combines two ideas, computing pseudo-GT from an optimization-based method and using it for distillation. When reading the paper, the idea would sound trivial but I think it's only after reading it. Prior to this paper, no previous work has presented this idea at least for the scene flow problem. One of the beauties of the paper is that the method is scalable. For scene flow (similar to optical flow), obtaining dense (pseudo) ground truth on real data is still a challenging problem. This paper presents an alternative way to obtain it and demonstrates that the trained model generalizes on real data and that it's scalable.
>
> I agree that the paper can look like a technical report. However, I personally like these in-depth analyses because they help understand why this scalable approach should work and how it behaves when scaling it up.

---

### Author Response · Authors · 2023-11-13

# Summary
We present a simple, effective scene flow method that leads to good performance (q16v, E12X). The paper features good writing, with clear motivation and implementation (E12X) and through experimentation (pGhB) that verifies the proposed idea (E12X) via in-depth analysis (q16v) that raises several intriguing questions with experiments to address these questions (pGhB).
# Reviewer Concerns
## Novelty

*Because our method is a simple but effective idea (q16v), E12X and pGhB raise concerns about novelty.*

Michael Black’s reviewer guidelines [1] eloquently argue that "the simplicity of an idea is often confused with a lack of novelty when exactly the opposite is often true. Such reviews lead researchers to make an idea appear more complex so that reviewers will find it of higher value."

Indeed, our novelty comes from the fact that our simple and post-hoc obvious pipeline produces state-of-the-art results. To quote q16v: "This distillation idea is very simple, but previous work on point-cloud-based scene flow methods hasn't tried the idea yet. The paper demonstrates that this simple idea is working and demonstrates that it's scalable by using large-scale diverse unlabeled data." This result also produces "valuable insights to the autonomous vehicle industry" (pGhB).

*E12X points out that much of our performance can be attributed to the components plugged into our Scene Flow via Distillation framework to create ZeroFlow.*

Our performance cannot be strictly attributed to the components plugged into our *Scene Flow via Distillation* framework — ZeroFlow outperforms both its unsupervised teacher, NSFP, and its supervised feedforward student, FastFlow3D.

But, broadly, we agree that ZeroFlow benefits from high quality components! Indeed, a feature of our *Scene Flow via Distillation* pipeline is, given a better test-time optimization method, a better supervised scene flow network architecture, or more unlabeled data, it can leverage these component improvements to produce an even better final method *for free*.

[1] Michael Black. Novelty in science: A guide to reviewers.

---

### Comment · Area_Chair_iY4a · 2023-11-22

Reviewers,

This is a reminder to reply to the rebuttal and post your final decisions TODAY.

AC

---

### Meta-Review · Area_Chair_iY4a · 2023-12-09

**Metareview:**

This paper describes "Scene Flow via Distillation" as a new, efficient framework for estimating 3D motion in point clouds, achieving top performance with no human labels and being faster and more cost-effective than current state-of-the-art methods. One reviewer recommends accepting the paper and the other two recommend rejection. The main concerns from the reviewers are the limited technical contributions of performing model distillation for scene flows. While acknowledging this, reviewer q16v still supports acceptance of the paper as it does address an important problem in scene flow, and the fact that it generalizes on real data and that it's scalable is interesting. After carefully reading the paper and the rebuttal and discussion. The AC recommends accepting the paper.

**Justification For Why Not Higher Score:**

While the paper presents good results, there are reservations from the reviewers on technical contributions.

**Justification For Why Not Lower Score:**

The results are solid.

---

### Decision · Program_Chairs · 2024-01-16

Accept (poster)